# On Stress-Induced Polarization Effect in Ammonothermally Grown GaN Crystals

**Karolina Grabianska \***[ID]**, Robert Kucharski, Tomasz Sochacki**[ID]**, Jan L. Weyher, Malgorzata Iwinska**[ID]**, Izabella Grzegory and Michal Bockowski**

Institute of High Pressure Physics, Polish Academy of Sciences, Sokolowska 29/37, 01-142 Warsaw, Poland; kucharski@unipress.waw.pl (R.K.); tsochacki@unipress.waw.pl (T.S.); weyher@unipress.waw.pl (J.L.W.); miwinska@unipress.waw.pl (M.I.); izabella.grzegory@unipress.waw.pl (I.G.); bocian@unipress.waw.pl (M.B.)
\* Correspondence: kgrabianska@unipress.waw.pl

**Abstract:** The results of basic ammonothermal crystallization of gallium nitride are described. The material is mainly analyzed in terms of the formation of stress (called stress-induced polarization effect) and defects (threading dislocations) appearing due to a stress relaxation process. Gallium nitride grown in different positions of the crystallization zone is examined in cross-polarized light. Interfaces between native ammonothermal seeds and new-grown gallium nitride layers are investigated in ultraviolet light. The etch pit densities in the seeds and the layers is determined and compared. Based on the obtained results a model of stress and defect formation is presented. New solutions for improving the structural quality of basic ammonothermal gallium nitride crystals are proposed.

**Keywords:** gallium nitride (GaN); basic ammonothermal growth method; interfaces; stress; etch pit density (EPD)

## 1. Introduction

No doubt, high structural quality gallium nitride (GaN) substrates are required for building a new generation of electronic and optoelectronic devices such as transistors or laser diodes [1,2]. One of the most promising technologies for crystallizing GaN is the ammonothermal method [3–12]. It involves dissolving the feedstock material (mainly GaN polycrystals) in ammonia (NH$_3$) in one zone of an ammonothermal autoclave and convective transport of the dissolved material to the second zone with native seeds where the crystal growth takes place. The crystallization process runs at NH$_3$ pressure of 300–600 MPa and temperature in the range of 300–750 °C [3]. In order to increase the feedstock solubility, some mineralizers such as alkali metals or halide compounds are applied. Two ammonothermal growth methods can be, respectively, distinguished: basic and acidic [13]. In the case of the basic ammonothermal crystallization process, GaN crystals (Am-GaN) of the highest structural quality can be obtained [14–17]. They are crystallographically flat with bowing radii of crystallographic planes higher than 15 m for a 2-inch crystal or wafer and with a low, of the order of $5 \times 10^4$ cm$^{-2}$, threading dislocation density (TDD). Depending on dopants and getters applied, highly conductive (n-type) or semi-insulating (SI) crystals can be grown. Two-inch Am-GaN substrates are available on the market [18].

In spite of the very high structural quality of the Am-GaN crystals, there is still room for their improvement. Recently, it was reported that the etch pit density (EPD; correlated with TDD) is two orders of magnitude lower in parts of crystals grown laterally (in the [11-20] crystallographic direction) than in the area grown in the [000-1] crystallographic direction [19]. Higher EPD in the central part of Am-GaN was combined with some stress existing in the crystals. This stress, visible in cross-polarized light, was called the stress-induced polarization effect (SIPE). It was stated that SIPE could appear on part or the entire surface of the crystal except for its laterally grown parts [15,19].

The process of the SIPE formation still remains an open question. It seems that it appears at the beginning of the growth process. At this point, the convective transport of reagents to the seeds can be non-uniform. As a result, the formation of a 3-dimensional (3D) growth mode and later hillocks and/or step bunching effect (due to constitutional supercooling) can take place on the crystallizing (000-1) plane. This, in turn, can create stress in the growing crystal, and therefore, SIPE will occur. Next, SIPE can lead to the formation of new dislocations (by a stress relaxation process). Finally, the TDD increases in crystals grown in the [000-1] crystallographic direction. Therefore, an increase in the EPD is observed.

In this paper, the formation of SIPE and its influence on the structural quality of Am-GaN are analyzed. Two selected crystals grown on seeds placed in different positions of an ammonothermal autoclave are examined. Interfaces between the native seeds and the new-grown Am-GaN are studied in detail in ultraviolet (UV) light illumination. Additionally, the EPD is determined and analyzed in the seed crystals and the new-grown Am-GaN layers.

## 2. Materials and Methods

### 2.1. General Remarks about Ammonothermal Crystal Growth Process

An ammonothermal crystallization process of GaN is conducted in a nickel-based alloy autoclave (reactor) of its own design and construction in a temperature range of 400–600 °C and $NH_3$ pressure of 0.3–0.4 GPa. A scheme of an ammonothermal autoclave with a typical interior configuration is presented in Figure 1a. Crucibles with GaN feedstock are arranged one above the other in the upper part of the reactor. In the lower part, GaN seeds are placed on special holders. These two zones are separated by a baffle.

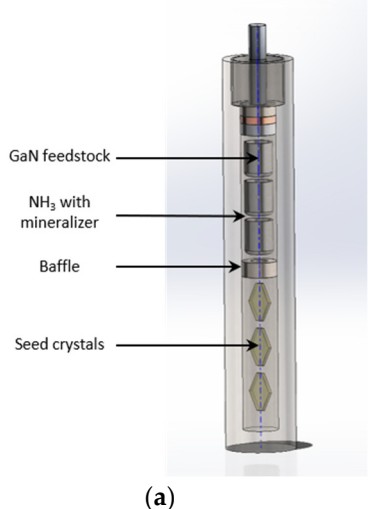

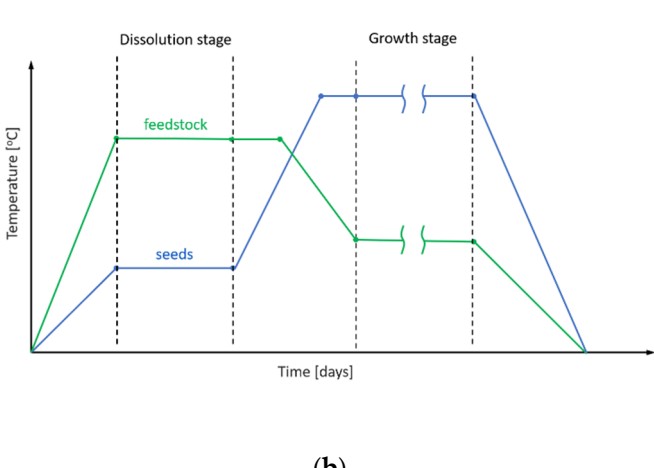

**(a)**            **(b)**

**Figure 1.** (**a**) Scheme of an ammonothermal autoclave with a typical interior configuration; (**b**) temperature–time profile of two zones in an autoclave: with feedstock and with seeds.

The seeds are always native ammonothermal GaN crystals. Their surfaces are not prepared in any special way. They can be as-grown, after slicing, after grinding, after mechanical polishing or epi-ready, i.e., after chemo-mechanical polishing (CMP). As already described [15,19], a typical ammonothermal crystallization run starts with a back-etching process. Then, all the seeds are kept at a lower temperature than the feedstock material. In time, the temperature profile changes, and the temperature of the growth zone with seed crystals is kept higher than that of the feedstock zone. The growth process starts, and GaN crystallizes on the etched surfaces of the native seeds. A scheme of the temperature–time relation of the two zones—with seeds and with feedstock—is presented in Figure 1b.

All the elements inside the autoclave are made of high-purity metals of well-known properties. Figure 2a presents typical metal holders with GaN seeds used for an am-

monothermal process. When the mounted seeds are of a diameter larger than 2 inches, they can be placed in four different positions. If the diameters of the seeds are smaller, they are placed close to each other, occupying an area for one 2-inch seed. Such a situation is marked by a black circle in Figure 2a. Generally, growth is conducted in the <000-1> as well as the lateral <11-20> crystallographic directions. The seeds are wrapped in metal foil and attached to the metal holder. Figure 2b shows a typical configuration of holders. Three holders can be placed at one level in the autoclave. There is space for 2 levels of holders. It allows for distinguishing eight positions (marked in Figure 2c) of the seed crystals mounted on the holders. Approximately 3550 crystals grown in these positions were analyzed. Figure 2d shows the average thicknesses of crystals grown on seeds placed in these 8 positions. It is clearly seen that these values vary for different seed positions. The crystals in the middle of the growth zone (Positions 3–6) are grown at a lower rate. It should also be noted that the highest percentage of SIPE presence (>50%) was recorded for Position 4.

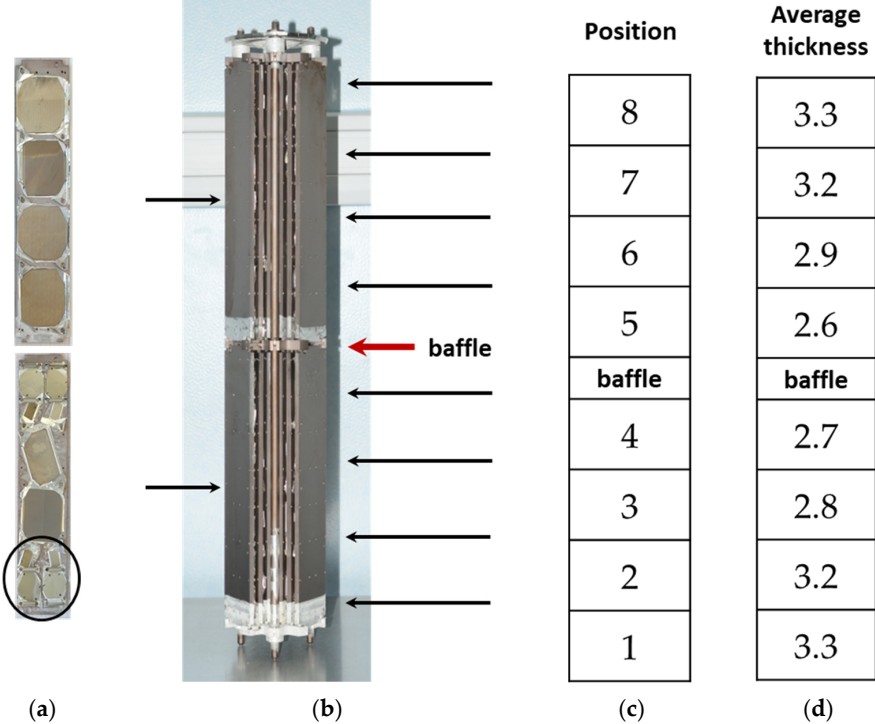

| Position | Average thickness |
|---|---|
| 8 | 3.3 |
| 7 | 3.2 |
| 6 | 2.9 |
| 5 | 2.6 |
| baffle | baffle |
| 4 | 2.7 |
| 3 | 2.8 |
| 2 | 3.2 |
| 1 | 3.3 |

(**a**)  (**b**)  (**c**)  (**d**)

**Figure 2.** (**a**) Typical metal holders with GaN seeds for ammonothermal growth run; black circle shows an example of filling the place of a 2-inch seed with a few smaller ones; (**b**) two levels of holders (one above the other) for placing inside the autoclave in the crystal growth zone; the levels are separated with a baffle; (**c**) scheme of eight positions (Positions 1–8) of seeds in the autoclave; (**d**) average thickness (in millimeters) of crystals grown at Positions 1–8.

### 2.2. Experimental

Figure 3 presents two rectangular seeds chosen for further growth process and examination. The (000-1) surface of each seed was prepared to an optically flat state by mechanical polishing with a 3 µm diamond grain gradation. No SIPE was detected in cross-polarized light in the two seeds. It means that no stress pattern was detected in them. Seed crystals No. 1 and No. 2 (see Figure 3) were mounted at Position 4 (Pos. 4) and Position 1 (Pos. 1), respectively, in the ammonothermal autoclave. A typical ammonothermal process, as described above, was performed, and it lasted 76 days.

After the crystallization run, the two chosen seeds and crystals grown on them were analyzed in cross-polarized and UV light. A polariscope and an optical microscope (OM) Nikon Eclipse LV100ND with Nomarski contrast and UV illumination were applied. The

(000-1) planes of the new-grown crystals, as well as cross-sections ((10-10)-planes) of the new-grown layer/seed systems, were examined in UV illumination. Attention was mostly paid to the interfaces. The samples were specially prepared for the UV illumination analyses (see Figure 4a). Without removing the seeds, some parts of crystals grown in the [000-1] direction (a few millimeters square) were diced and then mechanically polished in such a way that the new-grown layer, the interface and the seed's surface could be observed and examined at the same time. Figure 4b presents the way of preparing samples for analyzing EPD by the defect selective etching (DSE) method (for details, see [20]). The layers were sliced from the seeds above the interfaces. Next, the (0001) planes of the new-grown crystals as well as the seeds were prepared to an epi-ready state by chemo-mechanical polishing (CMP). These planes were then etched for 5 min in a KOH-NaOH solution at 450 °C. The samples were cleaned, and EPD was determined by counting etch pits at arbitrary chosen areas of a few square millimeters [21–23].

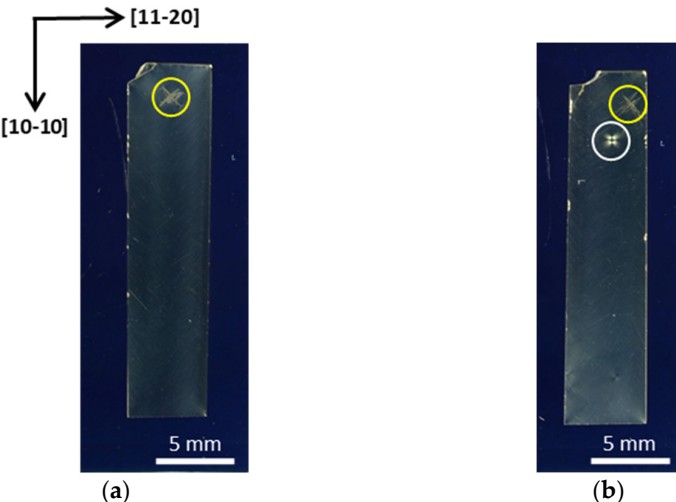

**Figure 3.** Two native rectangular Am-GaN crystals used as seeds: (**a**) No. 1 mounted at Pos. 4 of the autoclave; (**b**) No. 2 mounted at Pos. 1. of the autoclave; the white circle marked on crystal No. 2 shows the local stress area in the seed; crosses (marked with a pencil and surrounded with yellow circles) on the crystals allowed to distinguish the (000-1) and (0001) surfaces.

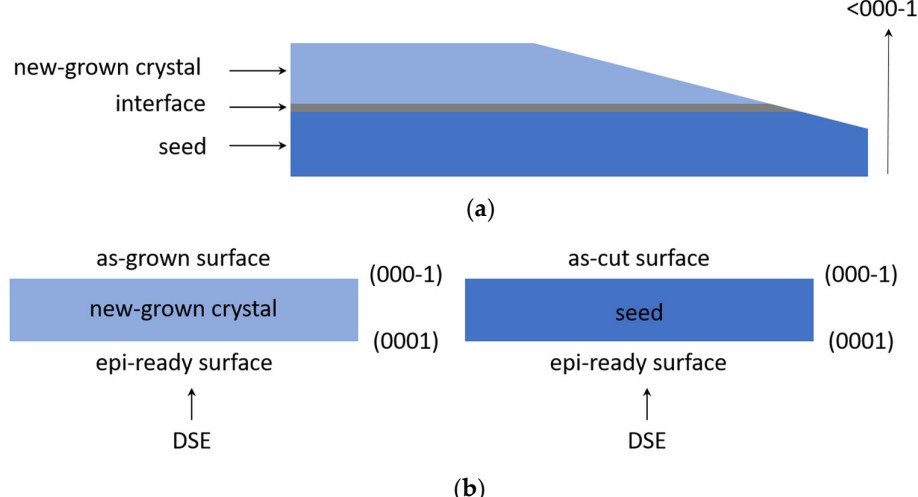

**Figure 4.** Schemes of: (**a**) cross-section of a sample prepared for analyzing new-grown crystal and seed with an interface between them under UV illumination; (**b**) new-grown crystal and seed prepared for DSE.

## 3. Results

Figure 5 shows the two analyzed crystals after the ammonothermal run (images in cross-polarized light). The average growth rates were 37 µm/day and 44 µm/day for Crystals No. 1 and No. 2, respectively. SIPE was found in crystal No. 1 grown at Pos. 4. No SIPE was observed for crystal No. 2 grown at Pos. 1. It is clearly visible that in both cases the parts grown in the lateral [11-20] crystallographic direction were free of SIPE.

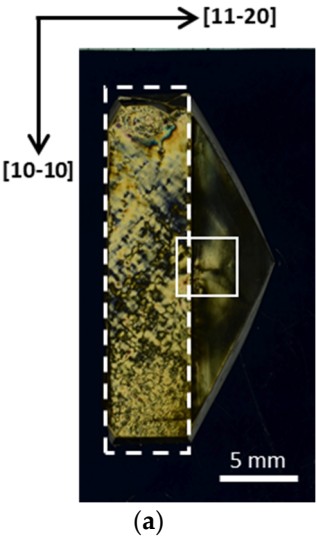
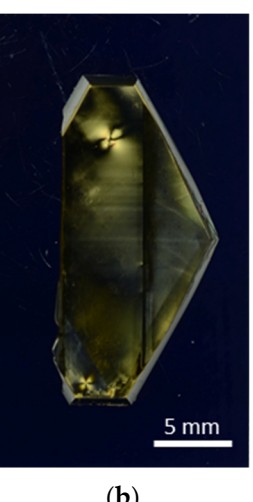

(a)                                    (b)

**Figure 5.** Two chosen crystals (see Figure 3) after the growth process; images in cross-polarized light: (**a**) crystal No. 1 (grown at Pos. 4), where the white square indicates a crack in the crystal formed during the growth process; in this case, SIPE is well visible in the crystal grown on the seed in the [000-1] direction (marked with a white dashed rectangle); part of new crystal grown in the lateral direction ([11-20]) is, in turn, SIPE-free; (**b**) crystal No. 2 (grown at Pos. 1); no SIPE is visible in crystal grown in the [000-1] and [11-20] directions; white circles show two local stress areas in the crystal.

Figure 6 presents the surfaces (in UV illumination) of the two analyzed samples prepared in the way described above and presented in Figure 4a. One can see the seed, the interface, and the new-grown crystal. For the sample where SIPE was detected (No. 1), some yellowish features are seen at the interface (see Figure 6a). Figure 6b demonstrates a magnification of one such feature at the interface (sample No. 1). A hexagonal symmetry of this yellowish figure is well visible. For crystal No. 2, without SIPE, the interface is flat with no visible features (see Figure 6c).

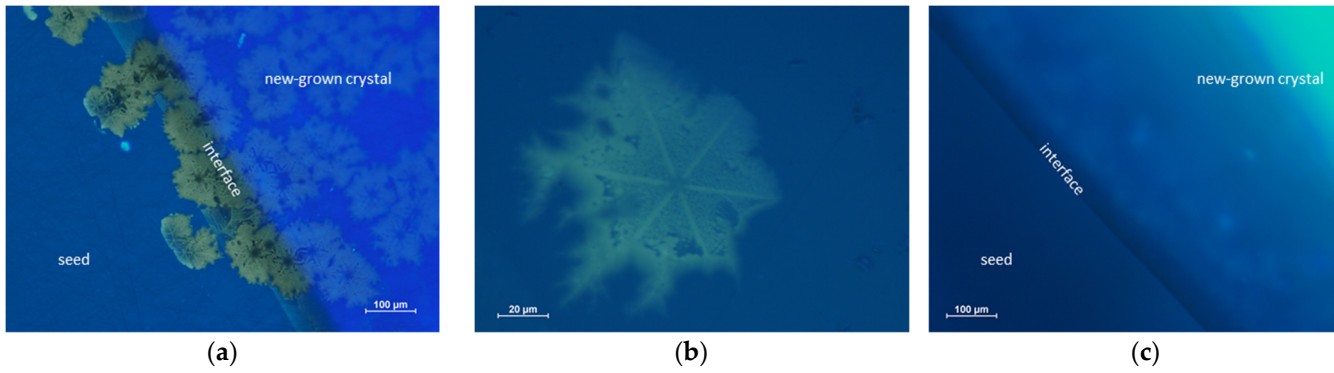

(a)                          (b)                          (c)

**Figure 6.** Images under UV illumination: (**a**) sample No. 1: seed, interface, and new-grown crystal; yellowish features are well visible at the interface; (**b**) magnification of one yellowish feature found at the interface (crystal No. 1); hexagonal symmetry should be noted; (**c**) sample No. 2: seed, interface, and new-grown crystal; no yellowish features visible at the interface.

Figure 7a shows a cross-section, the (10-10) plane, of crystal No. 1 at UV illumination. It is easy to distinguish between the seed and the new-grown crystal. The interface between them is wavy. Additionally, yellow luminescence (YL) is present. In crystal No. 2, in turn, the interface looked flatter, without any traces of YL (see Figure 7b).

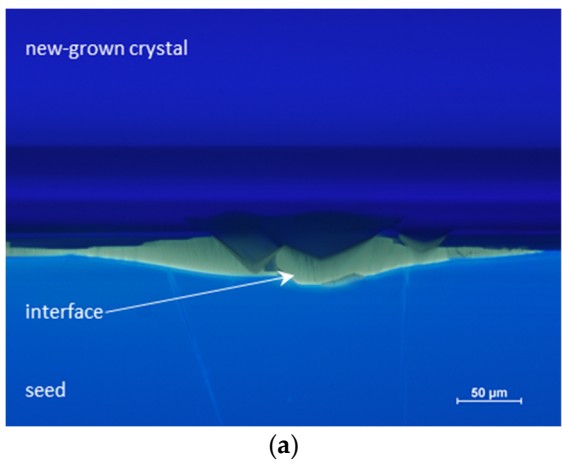
(**a**)

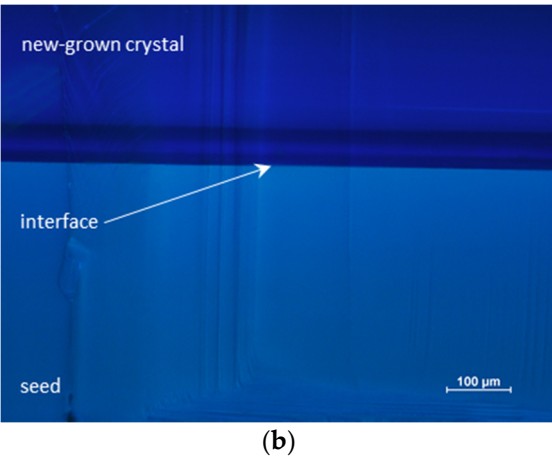
(**b**)

**Figure 7.** (**a**) Cross-section, the (10-10) plane, of the two examined crystals under UV illumination: (**a**) crystal No. 1; wavy interface between seed and new-grown crystal can be seen; YL at the interface is well visible; (**b**) crystal No. 2; flat interface without YL.

Figure 8 presents the seeds and the new-grown layers, prepared as presented in Figure 4b, after DSE. As can be seen, there is a big difference in the EPD (one order of magnitude; from $10^4$ cm$^{-2}$ to $3 \times 10^5$ cm$^{-2}$) between the seed and the new-grown layer for sample No. 1. For the second sample (No. 2), there is no difference in EPD in the seed and the new-grown crystal. For the same areas observed in the seed and the layer, even the positions of the pits look the same (see Figure 8b).

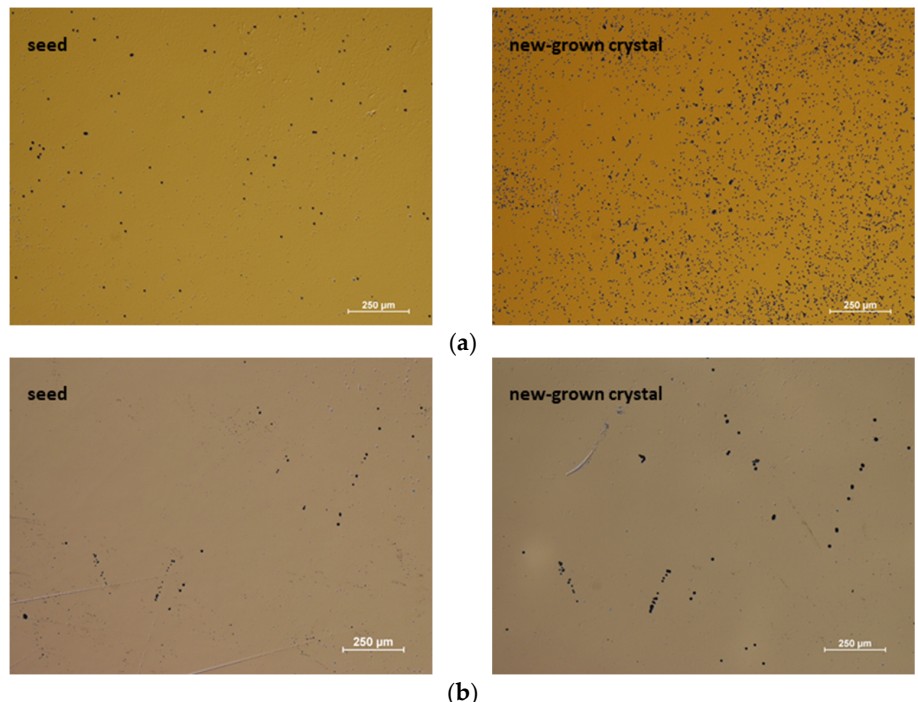

**Figure 8.** (0001) planes of seeds and new-grown crystals: (**a**) sample No. 1; (**b**) sample No. 2; only for crystal no. 1, there was a significant increase (one order of magnitude) in EPD in the new-grown crystal compared to the seed.

## 4. Discussion

The occurrence of SIPE has become the main factor inhibiting the proper development of basic ammonothermal crystallization. SIPE is the reason why TDD in Am-GaN crystals can occasionally be higher than $5 \times 10^4$ cm$^{-2}$. Eliminating SIPE allows improving the structural quality of the crystals and thus the substrates. As mentioned in the Introduction, SIPE can be associated with the beginning of the crystallization process and non-uniform supersaturation in the crystal growth zone and individual growth fronts of the crystals. As reported in [14,15,19], basic ammonothermal growth starts from the back-etching process. Then, the feedstock temperature is higher than the temperature of the crystal growth zone (see Figure 1). The seed crystals are dissolved in the solution. If the surface of the seed is non-uniform (after slicing, grinding, lapping, or mechanical polishing), the etching process will also be non-uniform across the surface of the seed. First of all, subsurface damage (with an extremely high concentration of point defects [24–27]) that always exists in a not epi-ready surface will be revealed. The waviness of the interface will be formed due to several factors, e.g., selectivity of the etching process (surface with more damage will be etched faster) and spreading of deep pits formed on threading structural defects.

In order to exclude the seed's surface preparation as the main reason for growth disturbance and SIPE formation, the (000-1) surfaces of the two seeds used for this research were in the same state: optically flat. Both seeds were free of SIPE. They were mounted at Positions 1 and 4 in the autoclave. As expected based on the statistics, the crystal grown at Pos. 4 demonstrated SIPE. The new crystal grown at Pos. 1 was SIPE-free. No doubt, the reason for SIPE formation was non-uniform supersaturation due to a non-uniform convective flow in the growth zone. In both cases, the growth started from an etched surface. Due to non-uniform convection, the etching was inhomogeneous, and, finally, the state of the (000-1) surface of the seeds at Positions 1 and 4 was different. At Pos. 4, a 3D growth mode appeared together with the formation of the yellowish features (see Figure 6a). The non-uniformly etched surface tended to minimize its free energy. Therefore, low-energy semi-polar planes evolved. This was accompanied by different incorporation of dopants (YL visible at the interface), resulting in a change in the lattice parameters [28–30]. The start of the growth process in the 3D mode caused the appearance of stress at the interface and then SIPE formation. This did not happen at Pos. 1, where the convection was more uniform, and the back-etching process and first stage of growth were quite homogeneous. In this case, the growth started in the two-dimensional (2D) growth mode with homogeneous incorporation of dopants. The stress was too low to be seen as SIPE. A similar phenomenon was also observed in crystal growth from the gas phase. Some impurities at the interface between the seed and the new-grown crystal can hinder or assist with later bulk growth [31,32].

The correctness of the presented model was confirmed by the DSE results and determination of EPD. When the interface between the seed and the new-grown crystal was flat but without significant incorporation of impurities (no YL; see Figure 7b), the value of EPD was the same in the new-grown layer and in the seed (see Figure 8b). This means that TDD did not increase in the new-grown crystal. When the interface was wavy with the presence of YL (significant incorporation of dopants), EPD increased in the layer. This indicates that TDD in the new-grown crystal also increased. The reason for this was the stress generated at the interface and then relaxation by the formation of new dislocations. The appearance of dislocations relaxes the crystal but never completely removes SIPE.

It should be mentioned that SIPE was not formed during the growth process in the lateral direction (see Figure 5). Crystallization in the [11-20] direction has to be realized in a different way than in the [000-1] direction. Growth begins at the edge of the seed crystal. There is no surface underneath the newly crystallized material. This presents a chance to improve the structural quality of basic ammonothermal crystals in the near future. New seeds with low TDD should be obtained by the lateral growth process. In parallel, the convective flow should be studied and then controlled in order to be uniform in all positions in the growth zone. This should be conducted for both the back-etching and

crystallization stages. The new seeds (without SIPE and with low TDD) could be used for the next generation crystallization processes in the [000-1] direction. Obviously, for completely excluding the 3D growth mode at the interface, the (000-1) surfaces of the seeds should have strictly defined misorientation (off-cut) and be prepared to an epi-ready state. Then, the back-etching conditions for the entire seed will be uniform. Thus, the surface will be homogenously etched (assuming that there are no cracks, inclusions, precipitates, or high TDD in the crystal). After such etching, the crystallization process should start from a 2D growth mode, by steps flown on the seed's surface. In time, due to a lack of the steps source, the (000-1) plane will be recovered, and the 2D growth mode will change. Some hillocks will be formed.

## 5. Conclusions

The formation of SIPE, as well as its consequence (the emergence of new defects), in ammonothermally grown GaN crystals was determined in detail. Through a careful surface and interface analysis, it was demonstrated that a non-uniform back-etching process and the start of the crystallization process in a 3D growth mode can lead to stress formation in the growing crystals. This happens due to the non-uniform incorporation of dopants at the interface between the native seed and the new-grown GaN. Stress relaxation leads, in turn, to the formation of new defects in the growing crystal. The lack of stress in the Am-GaN crystal (a potential seed) means that either the crystal is stress-free (or the existing stress is very low) or there is an increased density of threading dislocations (the stress has already relaxed).

It is generally assumed that the convective flow of reagents and the seed's surface preparation can influence the structural quality of the growing crystals and the formation of SIPE. In fact, the homogeneity of the growth process will depend on the uniformity of supersaturation. The latter, in turn, can be ensured by a homogenous and laminar convective flow of the reagents to the growth zone at proper and constant temperature–pressure conditions. A quantitative analysis of the convective transport at 3D approximation would allow us to find the best growth conditions for a uniform crystallization of GaN in each position in the crystal growth zone in an autoclave.

**Author Contributions:** K.G.—crystal growth experiments, characterization, preparing the manuscript; R.K.—crystal growth experiments, characterization; T.S.—crystal growth experiments, characterization; J.L.W.—DSE measurements; M.I.—review and editing; I.G.—supervision, review and editing; M.B.—supervision, review and editing. All authors have read and agreed to the published version of the manuscript.

**Funding:** This research was supported by the Polish National Science Center through Project No. 2021/41/N/ST5/03669 as well as by the Polish National Centre for Research and Development through project TECHMATSTRATEG-III/0003/2019-00. This research also received funding from the Department of the Navy, Office of Naval Research Global under ONRG award number N62909-21-1-2063 as well as the ECSEL Joint Undertaking (JU) under grant agreement No 101007310. The JU receives support from the European Union's Horizon 2020 research and innovation program and Italy, Germany, France, Poland, the Czech Republic, and the Netherlands.

**Institutional Review Board Statement:** Not applicable.

**Informed Consent Statement:** Not applicable.

**Data Availability Statement:** The data presented in this study are available on request from the corresponding author.

**Conflicts of Interest:** The authors declare no conflict of interest.

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
