# Peer review of "On Stress-Induced Polarization Effect in Ammonothermally Grown GaN Crystals"

_crystals, doi:10.3390/cryst12040554_

Round 1
Reviewer 1 Report
The authors reported a study on the stress-induced polarization effect in ammonothermally grown GaN crystals. Further, the Gallium nitride grown in different positions of the crystallization zone is examined, and etch pit density in the seeds and the layers are compared. Based on the results, new-grown gallium nitride layers exhibit better structural quality. The thesis of the manuscript is generally supported by the provided results. However, the following comments should be addressed before the recommendation for publication.
1. The authors mentioned in L104 that “No SIPE was detected in cross polarized light in the two seeds”. How did the authors confirm this information? Please explain it there for better understanding.
2. The quality of Figure 4. is poor. Can you please update with a better figure?
3. Can the authors explain the procedure clearly about the UV light illumination and imaging information?
4. How did the author calculate EPD? Please explain it.
5. The authors could not explain the growth process clearly and needs to be further improved. For better readability, can the authors explain it clearly with the help of a flow chart?
6. Many recent updates on the current study/topic are not covered in the introduction. Can the authors carefully revise the introduction with the required support of references?
Author Response
Drogi Recenzencie,
DziÄ™kujemy za komentarze i poprawki, które naprawdÄ™ pomogÅ‚y nam ulepszyć manuskrypt.

Reviewer 2 Report
The manuscript titled ''On stress induced polarization effect in ammonothermally grown GaN crystals'' co-authored by Grabianski et al. describes the stress induced polarization effect in the ammonothermal growth of GaN crystals. In my opinion, the manuscript is extremely important for the field of growth of III-nitride crystals and layers and in particular for the improvement of suitable GaN substrates with minimal dislocations which could immensely contribute to both basic research and industrial applications of III-nitrides. There are a few comments that I would like to state:
- NH3 is wrongly written i.e. the 3 should be in subscript. Please revise it for example in lines 29, 65.
- In line 47 SIPE is written as SPIE. This should be corrected.
- Quality of figure 4 must be improved.
- There are no studies of the microstructure using HRTEM and dislocation/strain analysis. I would like the authors to also make a thorough investigation of the grown crystals using both X-ray diffraction and high resolution transmission electron microscopy.
After these comments are taken care of, the manuscript with appropriate rebuttal must be submitted for further consideration.
Author Response
Dear Reviewer,
Thanks for your comments and corrections that really helped us improve the manuscript.

Reviewer 3 Report
This manuscript is dedicated to a study of ammonothermal crystallization process of GaN crystals (Am-GaN). The authors focus on solutions for improving the structural quality of ammonothermal gallium nitride crystals and on stress induced polarization effect. The work offers a sophisticated and well-planned experimental setup for ammonothermal crystallization processes and a deep analysis to achieve better understanding of the crystal growth. From practical point of view, the reported results have important bringing new knowledge about ammonothermal crystallization process GaN.
Thus, the ambitious task in this work covers an array of hot topics of research of the complex processes defining ammonothermal crystallization process of GaN with wide perspectives for groundbreaking applications impacting emerging group III nitride technologies that are currently attracting much research interest.
The authors chose an adequate structure of the manuscript – an excellent point of departure for such a study. Finally, the authors provided a balanced realistic and nicely illustrated presentation of their growth efforts and corresponding analysis that is of much scientific and practical interest and adds to new knowledge to the field.
In my opinion, the fine detailing in the present work, the insightful and balanced discussion of the results, as well as the very good figures, permit competent readers to utilize the manuscript as a guidance for future work. Consequently, this manuscript presents an efficient and beneficial basis for promoting and solving next step challenges in this field.
Moreover, the manuscript benefits from a clear motivation and it is an easy and informative read. The manuscript is also excellent in terms of clarity and accuracy of language.
The present manuscript is a significant contribution, this work once published would be quite useful as well as instructive and suggestive in terms of further studies and to a wider readership.
There are some minor issues with this already excellent manuscript that will need to be addressed before becoming suitable for publication, i.e., it can be considered for publication after a minor revision:
1: The authors miss part of bigger picture of crystal growth of III nitrides as assisted by . theoretical calculations and especially DFT methods, e.g., Journal of Physics D: Applied Physics 48 (2015) Article number 295104, Journal of Applied Physics 96 (2004) Pages 5293 - 5297. Such theoretical works treating defects/crack/dislocation formation on diversity of III nitrides, are supportive to the credibility of the experimental structural and chemical findings reported in the present manuscript.
2: The authors mention the temperature range 400-600°C together with NH3 pressure of 0.3-0.4 GPa. Thermal range may influence bonding/defects especially in relation to non-uniformity of seeds. It may be of practical to discuss this issue in more detail.
3: Authors mention “point” non-uniform incorporation of dopants. It is a hugely diversified topic in the growth of group III nitrides. It would be really informative if the authors discuss in more detail this aspect in the context of formation/density of acquired defects in the crystal.
4: Spell-check and stylistic revision of the paper are still necessary. Some long sentences, misspellings, etc., still are noticeable throughout the text.
Author Response

(The authors gave the same response as above.)

Round 2
Reviewer 1 Report
The authors have improved the quality of the manuscript and it can be accepted.
Reviewer 2 Report
I am satisfied with the revisions. I recommend publication of the manuscript